# DGQR estimation for interval censored quantile regression with varying-coefficient models

**ChunJing Li, Yun Li, Xue Ding, XiaoGang Dong** *

School of Mathematics and Statistics, Changchun University of Technology, Changchun, China

* dongxiaogang@ccut.edu.cn

**Data Availability Statement:** All relevant data are within the manuscript and its Supporting information files.

**Funding:** The author(s) received no specific funding for this work.

## Abstract

This paper propose a direct generalization quantile regression estimation method (DGQR estimation) for quantile regression with varying-coefficient models with interval censored data, which is a direct generalization for complete observed data. The consistency and asymptotic normality properties of the estimators are obtained. The proposed method has the advantage that does not require the censoring vectors to be identically distributed. The effectiveness of the method is verified by some simulation studies and a real data example.

## Introduction

Varying-coefficient models are among popular models that have been proposed to reduce the curse of dimensionality. They were natural extensions of classical parametric models and more popular in data analysis. Thanks to their flexibility and interpretability. Varying-coefficient models were frist introduced by Cleveland [1]. Hastie and Tibshirani [2] extended it to regression models and generalized regression models. Huang and Wu [3] proposed an inference program based on the resampling subject bootstrap, which is based on the varying-coefficient model. At present, there were many results of parameter estimation studies on quantile regression for varying-coefficient models, such as, Honda [4] considered varying-coefficient quantile regression. Cai and Xu [5] studied quantile regression estimation for varying coefficients dynamic models. Yuan and Ju [6] considered a varying-coefficient quantile regression model in which some covariates random missing, and proposed a weighted estimate based on empirical likelihood. Tang and Zhou [7] used inverse probability weighted method in the varying-coefficient composite quantile regression model with random missing covariates. Sun and Sun [8] proposed optimal inverse probability weighted estimation of regression parameters when selection probabilities were known in the quantile regression model with varying-coefficient.

We focus on the following varying-coefficient quantile regression model in this article:

$$Q_\tau(y_i|x_i, T_i) = x_i^\top \beta_\tau(T_i), \quad i = 1, \cdots, n.$$

where $\tau \in (0, 1)$, $y_i$ is the response variable of interest, which may represent the timing of the occurrence of some events, such as the time of death or disease, or some transformation of the time to the event [9], and $x_i$ is an observable covariate vector. $Q_\tau(y_i|x_i)$ is the conditional

**Competing interests:** The authors have declared that no competing interests exist.

quantile function [10] of $y_i$ given $x_i$, and $\beta_\tau(T) \in R^m$ is the coefficient function vector dependent on $\tau$.

However, in some practical applications, $y_i$ may not be fully observed due to the occurrence of censoring. For example, response variable $y_i$ is subjected to interval censoring: suppose one does not observe $y_i$, but censoring vector $t_{1i}$, $t_{2i}$, which satisfies $P(t_{1i} < y_i \leq t_{2i}) = 1$. Interval censored data is naturally produced in many clinical trials and longitudinal studies where individuals are tested regularly but not continuously. Interval censored data have been discussed by Sun [11] discuss several important topics about interval-censored failure time data that can occur in practice. Feng and Duan [12] studied a interval-censored data that distribution of or the underlying mechanisms behind censoring variables may depend on the treatment method, so it is different for subjects in different treatment groups. Chay and Powell [13], Ji and Peng [14], Li and Zhang [15], Lin and He [16], concerned linear regression with interval censored data. Zhou and Feng [17] propose an estimation method for quantile regression models with interval censored data. For varying-coefficient quantile regression model with censored data, Yin and Zeng [18] proposed a varying-coefficient quantile regression model subject to random censoring. Xie and Zhou [19] adopted a weighted inverse probability approach to develop a varying-coefficient model to the estimation of regression quantiles under random data censoring. These studies have not considered the coefficient function estimation method of the interval censored data.

The primary goal of this article is to develop a estimate method with interval censored data. We will use methods to estimate the coefficient function vector $\beta_\tau(T)$ for general $\tau \in (0, 1)$. We propose a direct generalization quantile regression(DGQR) estimation method and first to develop theory and methodology of the quantile regression for varying-coefficient models with interval censored data. Under some regularity conditions, obtain the asymptotic normality of $\hat{\beta}_\tau(t)$. The proposed estimator is defined as the optimal solution point of a minimization problem with convex objective function. The property of asymptotic normality is established with a bias converging to zero. We also compared the performance of our proposed method with other methods in the quantile regression with varying-coefficient models.

The rest of this paper is arranged as follows. In Section 2, we put forward the DGQR estimation method to quantile regression for varying-coefficient model with interval censored response observations. In Section 3, establish asymptotic properties of the estimator. In Section 4, simulations are achieved to investigate the finite sample performance of the proposed methods, and simulation results show that the proposed methods work well for various $\tau \in (0, 1)$. Section 5 gives an example analysis. A conclusion are given in Section 6. In the appendix to Section 7, technical proofs are given.

## DGQR estimation

We consider the following varying-coefficient model:

$$Y = X^\top \beta(T) + \varepsilon, \tag{1}$$

where $Y \in R$ is a response variable, $X = (X_1, \cdots, X_p)^\top \in R^p$ is a $p$-dimensional covariate, $\beta(\cdot) = (\beta_1(\cdot), \cdots, \beta_p(\cdot))^\top$ is an unknown vector-valued function with a smoothing variable $T$, the components $\beta_j(\cdot)$ ($j = 1, 2, \cdots, p$) are all differentiable functions, $\varepsilon$ is the random error whose $\tau$th quantile is zero, i.e.,

$$\int_{-\infty}^{0} f(\varepsilon) d\varepsilon = \tau,$$

where $f(\varepsilon)$ denotes the density function of $\varepsilon$. $\varepsilon$ is also assumed to be independent with $X$ and $T$.

In what follows, we first briefly introduce the quantile regression (QR) estimates under complete data. Then we discuss in detail the quantile regression method under the interval censored data. Throughout the paper, we denote $\beta'(t)$ the derivative function of $\beta(t)$. Denote $\|\cdot\|$ the $L_2$ norm of the corresponding vector.

Note that $\beta_j(T)$ is differentiable. By Taylor's expansion, we have [7]

$$\beta_j(T) \approx \beta_j(t) + \beta_j'(t)(T - t) := a_j + b_j(T - t), \ j = 1, \cdots, p.$$

Thus, if all data $\{y_i\}_{i=1}^n$ are observable, the QR estimator $\tilde{\beta}(t)$ of $\beta(t)$ [4] is defined as

$$\tilde{\beta}_n(t) = \arg\min\left\{\sum_{i=1}^n \rho_\tau(y_i - x_i^\top[a + b(T_i - t)])K\left(\frac{T_i - t}{h}\right)\right\},$$

for some fixed $\tau \in (0, 1)$, where $a = (a_1, \cdots, a_p)^\top$, $b = (b_1, \cdots, b_p)^\top$, $K\left(\frac{T_i-t}{h}\right)$ is a kernel function with bandwidth $h$, $\rho_\tau(s) = s(\tau - I(s < 0))$ is the loss function (see, e.g., Koenker (2001) [20].), i.e.,

$$\rho_\tau(y_i - x_i^\top[a + b(T_i - t)]) = \begin{cases} \tau|y_i - x_i^\top[a + b(T_i - t)]|, & y_i \geq x_i^\top[a + b(T_i - t)]; \\ (1-\tau)|y_i - x_i^\top[a + b(T_i - t)]|, & y_i < x_i^\top[a + b(T_i - t)]. \end{cases}$$

Next, we focus on the interval censoring case, i.e., $y_i$ can not be observed, and we can only observe two point $t_{1i}$ and $t_{2i}$ satisfying $t_{1i} < y_i \leq t_{2i}$. Suppose the length of interval $t_{2i} - t_{1i}$ is small. Then $y_i$ will be close to $t_{1i}$ and $t_{2i}$. Under this assumption and some other regularity conditions, the probability of $P(x_i^\top[a + b(T_i - t)] \in (t_{1i}, t_{2i}])$ will be close to zero. Thereby, we can modify the loss function $\rho_\tau(y_i - x_i^\top[a + b(T_i - t)])$ by using the method proposed by Zhou and Feng [17]. Define this method as DGQR estimation, i.e.,

$$F_\tau(t_{1i}, t_{2i}, x_i^\top[a + b(T_i - t)]) = \begin{cases} \tau|t_{1i} - x_i^\top[a + b(T_i - t)]| & , x_i^\top[a + b(T_i - t)] \leq t_{1i}; \\ 0 & , t_{1i} < x_i^\top[a + b(T_i - t)] \leq t_{2i}; \ (2) \\ (1-\tau)|t_{2i} - x_i^\top[a + b(T_i - t)]| & , t_{2i} < x_i^\top[a + b(T_i - t)]. \end{cases}$$

In 2), we use $F_\tau(\cdot)$ instead of $\rho_\tau(\cdot)$ to make the notation clearer. Based on 2), the DGQR estimator $\hat{\beta}_n(t)$ for interval censored varying-coefficient model 1) can be obtained by minimizing the following criterion function

$$\min_{\theta\in\Theta}\left\{\sum_{i=1}^n F_\tau\left(t_{1i}, t_{2i}, x_i^\top[a + b(T_i - t)]\right)K\left(\frac{T_i - t}{h}\right)\right\}, \tag{3}$$

i.e.,

$$\hat{\beta}_n(t) = \arg\min_{\theta\in\Theta}\left\{\sum_{i=1}^n F_\tau\left(t_{1i}, t_{2i}, x_i^\top[a + b(T_i - t)]\right)K\left(\frac{T_i - t}{h}\right)\right\}. \tag{4}$$

Obviously, if $y_i$ are exactly observed, i.e. $t_{1i} = t_{2i}$ holds for each $i$, the DGQR estimator $\hat{\beta}_n(t)$ defined in (4) will be reduced to quantile estimator $\tilde{\beta}_n(t)$ for the complete observed data.

## Asymptotic properties

To study the asymptotic properties of varying-coefficient DGQR estimator $\hat{\beta}_n(t)$, we first give some assumptions.

C.1. The density function $f(\cdot)$ of $\varepsilon$ has a continuous and uniformly bounded derivative, namely $0 < \sup_s f'(s) < B_0$.

C.2. $(x_1^\top, t_{11}, t_{21}), \cdots, (x_n^\top, t_{1n}, t_{2n})$ are the independent and identically distributed (i.i.d.) sample from random vector $(X_i^\top, t_{1i}, t_{2i})$ which is subject to the condition in Lemma 2.

C.3. Matrix $E(X_i X_i^\top)$ is a positive definite matrix, and $E(X_i) = 0$.

C.4. Random variable $T$ has a second-order differentiable density function $f_T(t) > 0$ in some neighborhood of $t$ [7].

C.5. The kernel function $K(\cdot)$ is a symmetric density function with a compact support, whose bandwidth $h \to 0$, $nh \to \infty$ as $n \to \infty$ [7].

C.6. $(t_{1i}, t_{2i})(i = 1, \cdots, n)$ are independent random vectors (not necessary to be indentically distributed) which satisfy $sup_i |t_{2i} - t_{1i}| \le \varrho_n$ for some sequence of $\varrho_n \to 0$ as $n \to 0$. Moreover, $G_i^1(\cdot)$ and $G_i^2(\cdot)$ are the marginal distribution functions of $t_{1i}$ and $t_{2i}$, which has continuous and bounded dervatives at the point $x_i^\top \beta(T_i) - r_i(t)$.

C.7. For each $\epsilon > 0$, there is a finite $M$ satifying

$$E\left[\frac{1}{n}\sum_{i=1}^{n}\|x_i\|^2 I(\|x_i\| > M)\right] < \epsilon,$$

which holds for all $n$ large enough.

C.8. The sequence of the smallest eigenvalues of the matrices

$$H_n = E\left\{\frac{1}{n}\sum_{i=1}^{n} x_i^* x_i^{*\top}\left[(1-\tau)\frac{\partial G_i^2(\ell)}{\partial \ell}\Big|_{\ell=x_i^\top \beta(T_i)} + \tau\frac{\partial G_i^1(r)}{\partial r}\Big|_{r=x_i^\top \beta(T_i)}\right]\right\},$$

is bounded away from zero for some $n$ large enough, where $x_i^* = (x_i^\top, x_i^\top (T_i - t)/h)^T$.

Now we are ready to state the consistency and asymptotic normality of the QR estimators $\hat{\beta}_n(t)$.

**Theorem 1**. For any $\tau \in (0, 1)$, under Assumptions C.1-C.8,

$$\hat{\beta}(t) \xrightarrow{P} \beta_0(t),$$

holds as $n \to +\infty$, where "$\xrightarrow{P}$" stands for convergence in probability, and $\hat{\beta}(t) = (\hat{a}^\top, \hat{b}^\top)^\top$, $\beta_0(t) = (\beta(t), \beta'(t))$.

**Theorem 2**. For $\tau \in (0, 1)$, under Assumptions C.1-C.8,

$$\tilde{H}_n^{-1/2}\left(\sqrt{nh}H_n f_T(t)\begin{pmatrix}(\hat{a} - \beta(t))^\top \\ (\hat{b} - \beta'(t))^\top h\end{pmatrix} + L_n\right) \xrightarrow{d} N(0, E_m),$$

holds as $n \to +\infty$, where $E_m$ denotes the identity matrix of order $m$, "$\xrightarrow{d}$" stands for

convergence in distribution, and

$$\tilde{H}_n = \frac{1}{n}\sum_{i=1}^{n} E x_i^*(x_i^*)^\top [\tau^2 P_{1i} + (1-\tau)P_{2i} + 2\tau(\tau-1)P_i] f_T(t),$$

$$P_{1i} = P(x_i^\top \beta(T_i) \le t_{1i}|X_i, T_i)P(x_i^\top \beta(T_i) > t_{1i}|X_i, T_i) + o(1),$$

$$P_{2i} = P(x_i^\top \beta(T_i) > t_{2i}|X_i, T_i)P(x_i^\top \beta(T_i) \le t_{2i}|X_i, T_i) + o(1),$$

$$P_i = P(x_i^\top \beta(T_i) > t_{2i}|X_i, T_i)P(x_i^\top \beta(T_i) \le t_{1i}|X_i, T_i) + o(1),$$

$$L_n = \frac{1}{\sqrt{nh}}\sum_{i=1}^{n} E\{x_i^*[(1-\tau)I(x_i^\top \beta(T_i) - r_i(t) > t_{2i}) - \tau I(x_i^\top \beta(T_i) - r_i(t) \le t_{1i})]K_i(t)\}.$$

## Simulations

In all simulations, we always use the Uniform kernel [21], that is $K(t) = \frac{1}{2}I\left(\left|\frac{T-t}{h}\right| \le 1\right)$, and use the bandwidths $h = 0.5n^{-1/3}$. For each scenario, we report the BIAS and mean-squared error (MSE) of parameter estimators based on 500 replications, which is defined as

$$BIAS = \frac{1}{n}\sum_{j=1}^{n}\left\{\hat{\beta}(t_j) - \beta(t_j)\right\},$$

$$MSE = \frac{1}{n}\sum_{j=1}^{n}\left\{\hat{\beta}(t_j) - \beta(t_j)\right\}^2.$$

**Example 1**. In this example, we adopt a data generation process similar to Kim *et al* [22]. With the regression model $y_i = x_i^\top \beta(T_i) + \varepsilon_i$ where coefficient function is $\beta(T_i) = T_i$, the observed data $\{(t_{1i}, t_{2i}, x_i, T_i)\}$ are generated as follows:

(1) Sample covariate $\{x_i\}$ from a standard normal distribution with Normal(0,1).

(2) Generate $\{T_i\}$ from Uniform(0.9,1.1).

(3) For each $i$, to generate censoring interval $(t_{1i}, t_{2i}]$, firstly we let $u_i = \min\{y_i\} - 0.3 + r_i$, with $r_i \sim Uniform(0, 0.3)$. Then choose $(u_i + \sum_{j=0}^{k-1} l_j, u_i + \sum_{j=0}^{k} l_j)$ as $(t_{1i}, t_{2i})$, where $l_0 = 0$, $l_j$ is generated from Uniform(0,0.3) independently for $j = 1, \cdots, k$, and $k$ is a non negative integer which satisfies $u_i + \sum_{j=0}^{k-1} l_j < y_i \le u_i + \sum_{j=0}^{k} l_j$.

(4) $\{\varepsilon_i\}$ are generated independently from the following four distributions:(a) Normal (0,0.1); (b) Logistic(0,0.3); (c) Lognormal(0,0.1); (d) Weibull(2.0,1.0).

Since the method proposed by Zhou and Feng [17] (Zhou estimation) can also be directly applied to quantile regression with varying-coefficient models. We are mainly interested in comparing the performance of the method proposed by Zhou and Feng [17] and ours (DGQR) in the quantile regression with varying-coefficient models. Frist we do simulations to compare these two methods for models with $\tau = 0.5$ and sample size $n = 200$. The simulation results of quantile regression with varying-coefficient models, Zhou estimation, and DGQR estimation, including BIAS and MSE, are presented in Table 1.

**Table 1. BIAS and MSE of two methods simulation results for Example 1.**

| $e_i$ | Method | BIAS | MSE |
|---|---|---|---|
| Normal(0, 0.1) | DGQR | 0.0004 | 0.0002 |
| | Zhou | −0.0007 | 0.0034 |
| Logistic(0, 0.2) | DGQR | 0.0011 | 0.0021 |
| | Zhou | −0.0011 | 0.0041 |
| Lognormal(0, 0.3) | DGQR | 0.0007 | 0.0017 |
| | Zhou | 0.0009 | 0.0041 |
| Weibull(3.0, 1.0) | DGQR | 0.0016 | 0.0022 |
| | Zhou | −0.0011 | 0.0040 |

**Example 2**. The performance of the proposed method for interval censored quantile regression with varying-coefficient models with different $\tau \in (0, 1)$, generate random data $\{(t_{1i}, t_{2i}, x_i\}$ from the same models as in Example 1 except that coefficient function is $\beta(T_i) = \sin(2\pi T_i)$ and $\{T_i\}$ from Uniform(0,1). We focus on comparing the BIAS and MSE(in brackets) with sample size $n$ = 100, 200 and 300. Then calculation BIAS and MSE of varying-coefficient models for $\tau$ takes four different values: 0.2, 0.4, 0.6, 0.8.

**Example 3**. We generate random data $\{(t_{1i}, t_{2i}, x_i, T_i)\}$ from the same models as in Example 2 except that coefficient function is $\beta(T_i) = 2T^2 + 6T$, and calculat BIAS and MSE for $\tau$ takes four different values: 0.2, 0.4, 0.6, 0.8.

**Example 4**. We generate random data $\{(t_{1i}, t_{2i}, x_i, T_i)\}$ from the same models as in Example 2 except that $\{x_i\}$ are derived independently from the distribution Exp(1), and calculat BIAS and MSE for $\tau$ takes four different values: 0.2, 0.4, 0.6, 0.8.

We summarize our findings below:

(1) From Table 1, we can see that the estimation method (DGQR) we proposed in terms of BIAS and MSE is superior than the method proposed by Zhou and feng [17], for the quantile regression for varying-coefficient models.

(2) As is seen in Tables 2–4, all the biases and MSE decrease as $n$ increases with different values of $\tau$, the estimates seem to be unbiased. This implies our estimates are consistent for all the parameters.

**Table 2. BIAS and MSE (in parentheses) of four distribution simulation result for Example 2.**

| $n$ | $\tau$ | Normal(0, 0.1) | Logistic(0, 0.2) | Lognormal(0, 0.3) | Weibull(3.0, 1.0) |
|---|---|---|---|---|---|
| 100 | 0.2 | 0.0012 (0.0038) | −0.0032 (0.0237) | −0072 (0.0133) | −0.0062 (0.0207) |
| | 0.4 | 0.0028 (0.0038) | −0.0002 (0.0163) | −0.0012 (0.0124) | −0.0012 (0.0156) |
| | 0.6 | 0.0062 (0.0039) | −0.0041 (0.0169) | −0.0041 (0.0139) | −0.0027 (0.0199) |
| | 0.8 | −0.0060 (0.0044) | −0.0127 (0.0272) | 0.0048 (0.0251) | −0.0030 (0.0223) |
| 200 | 0.2 | 0.0034 (0.0015) | 0.0039 (0.0125) | −0.0040 (0.0069) | −0.0061 (0.0095) |
| | 0.4 | −0.0012 (0.0013) | 0.0002 (0.0078) | 0.0015 (0.0059) | 0.0007 (0.0077) |
| | 0.6 | −0.0045 (0.0013) | 0.0352 (0.0104) | −0.0034 (0.0083) | −0.0046 (0.0083) |
| | 0.8 | −0.0134 (0.0017) | −0.0350 (0.0128) | −0.0048 (0.0134) | 0.0013 (0.0121) |
| 300 | 0.2 | 0.0048 (0.0010) | 0.0109 (0.0045) | 0.0172 (0.0026) | 0.0165 (0.0038) |
| | 0.4 | −0.0006 (0.0008) | −0.0015 (0.0030) | 0.0046 (0.0024) | −0.0039 (0.0030) |
| | 0.6 | −0.0097 (0.0010) | −0.0117 (0.0031) | −0.0082 (0.0031) | −0.0120 (0.0034) |
| | 0.8 | −0.0118 (0.0011) | −0.0230 (0.0049) | −0.0207 (0.0054) | −0.0258 (0.0045) |

**Table 3. BIAS and MSE (in parentheses) of four distribution simulation result for Example 3.**

| $n$ | $\tau$ | Normal(0, 0.1) | Logistic(0, 0.2) | Lognormal(0, 0.3) | Weibull(3.0, 1.0) |
|---|---|---|---|---|---|
| 100 | 0.2 | 0.0027 (0.0012) | 0.0158 (0.0306) | 0.0077 (0.0028) | −0.0971 (0.0301) |
| | 0.4 | 0.0034 (0.0013) | 0.0945 (0.0245) | 0.0191 (0.0031) | 0.0964 (0.0379) |
| | 0.6 | 0.0004 (0.0014) | 0.0914 (0.0249) | −0.0163 (0.0012) | 0.0621 (0.0283) |
| | 0.8 | −0.0100 (0.0013) | 0.0512 (0.0547) | 0.0405 (0.0029) | 0.0612 (0.0584) |
| 200 | 0.2 | 0.0153 (0.0008) | −0.0058 (0.0218) | 0.0067 (0.0007) | 0.0387 (0.0144) |
| | 0.4 | −0.0109 (0.0006) | −0.0045 (0.0114) | −0.0034 (0.0003) | −0.0204 (0.0091) |
| | 0.6 | −0.0182 (0.0006) | 0.0506 (0.0196) | 0.0047 (0.0007) | 0.0731 (0.0163) |
| | 0.8 | −0.0205 (0.0006) | 0.0076 (0.0048) | −0.0012 (0.0020) | −0.0020 (0.0120) |
| 300 | 0.2 | −0.0004 (0.0004) | 0.0237 (0.0149) | 0.0016 (0.0005) | −0.0103 (0.0059) |
| | 0.4 | 0.0044 (0.0004) | 0.0523 (0.0090) | 0.0022 (0.0002) | 0.0195 (0.0027) |
| | 0.6 | 0.0057 (0.0003) | −0.0381 (0.0081) | 0.0180 (0.0007) | −0.0204 (0.0095) |
| | 0.8 | −0.0111 (0.0002) | 0.0121 (0.0073) | −0.0003 (0.0007) | 0.0514 (0.0096) |

(3) Table 2 shows the BIAS and MSE of different residual distributions under the parameter settings of Example 2. We see that the values of bias do not differ much from their corresponding MSE, indicating that the estimators converge fast. Compared with Tables 2 to 4, all simulation result performs well, regardless the distrubution type of the covariates and the coefficients.

(4) Figs 1 and 2 show the DGQR estimator $\hat{\beta}(t)$ based on the Example 2 and Example 3 in the case of $\tau = 0.5$, respectively. From Figs 1 and 2, we can see that the biases of the estimator $\hat{\beta}(t)$ is very small. This further confirms that our proposed estimation method is effective.

## Empirical analysis

In this section, we will use the proposed DGQR estimation and interval generation mechanism procedure to analyze the air pollution data set collected by the Norwegian Public Roads Administration. The data set consists of 500 observations and can be found in StatLib. The data includes the concentration of $NO_2(y_i)$ per hour of the day, the number of cars per hour ($x_{1i}$), the wind speed ($x_{2i}$) and the hour ($T_i$). We use varying-coefficient model based quantile

**Table 4. BIAS and MSE (in parentheses) of four distribution simulation result for Example 4.**

| $n$ | $\tau$ | Normal(0, 0.1) | Logistic(0, 0.2) | Lognormal(0, 0.3) | Weibull(3.0, 1.0) |
|---|---|---|---|---|---|
| 100 | 0.2 | 0.0055 (0.0050) | −0.1067 (0.1130) | −0.0062 (0.0054) | 0.0923 (0.0201) |
| | 0.4 | 0.0026 (0.0033) | −0.0974 (0.0246) | 0.0002 (0.0044) | 0.0181 (0.0169) |
| | 0.6 | −0.0174 (0.0024) | −0.0405 (0.0137) | −0.0065 (0.0040) | −0.0907 (0.0195) |
| | 0.8 | −0.0234 (0.0050) | −0.0546 (0.0373) | 0.0071 (0.0031) | −0.0756 (0.0164) |
| 200 | 0.2 | 0.0058 (0.0015) | 0.0251 (0.0145) | 0.0128 (0.0019) | 0.0227 (0.0061) |
| | 0.4 | −0.0133 (0.0018) | 0.0293 (0.0149) | 0.0008 (0.0012) | 0.0233 (0.0068) |
| | 0.6 | −0.0024 (0.0011) | −0.0278 (0.0058) | −0.0032 (0.0013) | 0.0108 (0.0052) |
| | 0.8 | −0.0160 (0.0014) | −0.0184 (0.0129) | −0.0049 (0.0019) | −0.0660 (0.0090) |
| 300 | 0.2 | 0.0123 (0.0009) | 0.0321 (0.0080) | −0.0022 (0.0008) | 0.0044 (0.0028) |
| | 0.4 | −0.0062 (0.0009) | −0.0261 (0.0081) | −0.0002 (0.0008) | −0.0010 (0.0036) |
| | 0.6 | −0.0146 (0.0008) | −0.0272 (0.0044) | 0.0024 (0.0004) | 0.0055 (0.0027) |
| | 0.8 | −0.0175 (0.0008) | −0.0680 (0.0098) | −0.0127 (0.0007) | −0.0560 (0.0085) |

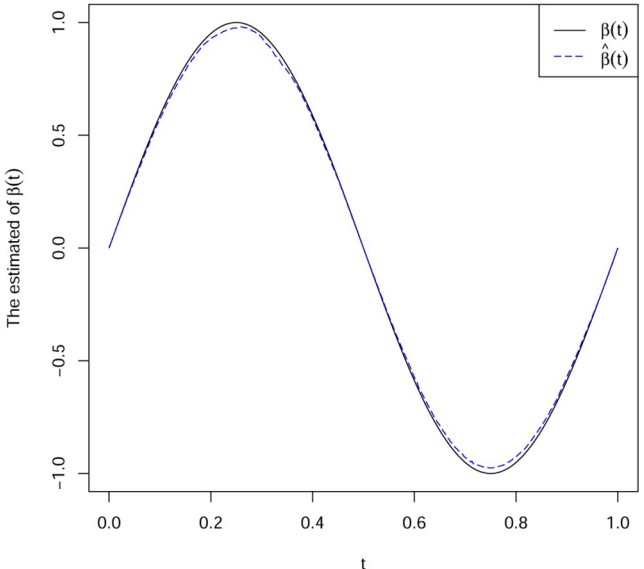

**Fig 1. Parameter setting based on Example 2 and $\tau$ = 0.5.** The solid curves true function $\beta$(t); dotted line estimated function $\hat{\beta}(t)$.

regression method to fit the data. We establish the following varying-coefficients model:

$$y_i = \log\left(x_{1i}\right)^\top \beta_1\left(T_i\right) + \log\left(x_{2i}\right)^\top \beta_2\left(T_i\right) + \varepsilon_i. \tag{5}$$

We use the interval generation mechanism in the simulation which generates interval $(t_{1i}, t_{2i}]$ with $y_i$.

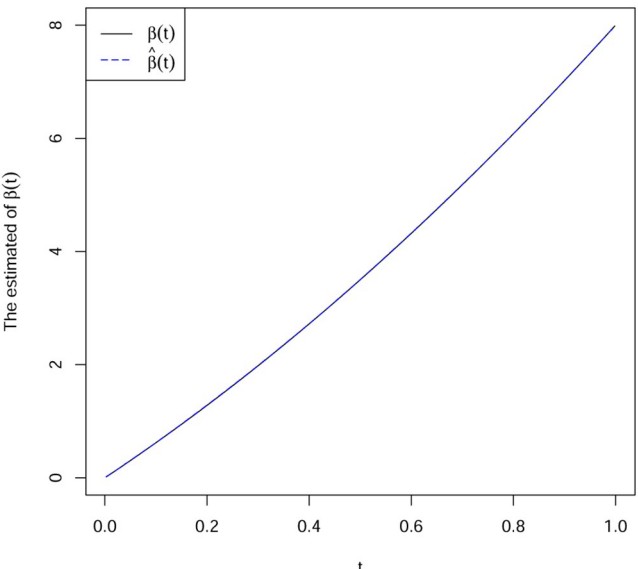

**Fig 2. Parameter setting based on Example 2 and $\tau$ = 0.5.** The solid curves true function $\beta$(t); dotted line estimated function $\hat{\beta}(t)$.

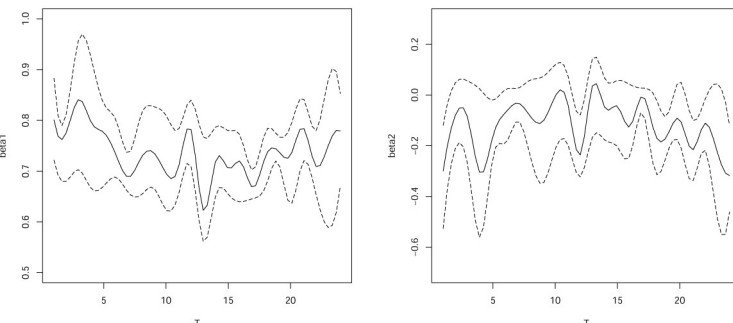

**Fig 3. Estimates and the corresponding pointwise confidence interval of $\beta 1(t)$, $\beta 2(t)$ for complete data.**

In order to test whether the coefficient function really time varying, we consider the following test questions:

$$H_0 : \beta(T_i) = \beta \ VS \ H_1 : \beta(T_i) \neq \beta,$$

where $\beta = c(\beta_1, \beta_2)$ is a constant vector. Based on 200 bootstrap resampling, we analyze interval censored data and give estimated functions of $\beta_1(T)$ and $\beta_2(T)$, along with the 95% bootstrap confidence bands, respectively. The p-values of test $T_n$ are both 0.00. Therefore, we should reject null hypothesis $H_0$ at a significance level of 0.05. Prove that model (5) is a varying-coefficient model.

Fig 3 plots the confidence intervals for $\beta_1(T)$ and $\beta_2(T)$ of the quantile regression for varying-coefficient models with completed data. Fig 4 plots the confidence intervals for $\beta_1(T)$ and $\beta_2(T)$ with interval censored data. The result in Fig 3 show that $\beta_1(T)$ and $\beta_2(T)$ are significant time varying with completed data and Fig 4 also show that $\beta_1(T)$ and $\beta_2(T)$ are significant time varying with interval censored data. Furthermore, we can also see that the DGQR estimators confidence intervals with the completed data as long as with the interval censored data. Basically, we can see that $\beta_1(T)$ and $\beta_2(T)$ of completed data and interval censored data the results are consistent in the confidence interval. And there is no loss effect.

To further illustrate the effect of fitting, we perform the following residual analysis. Fig 5 plots the residual histogram (a) and AFC plot (b) of the model fitted to the data. We can see the residual histogram plot(a) it is close to the normal distribution, and the residual sequence cannot be seen to be correlated in the corresponding AFC chart (b). This fitting result also confirms the advantage of the varying-coefficient quantile model in fitting interval censored

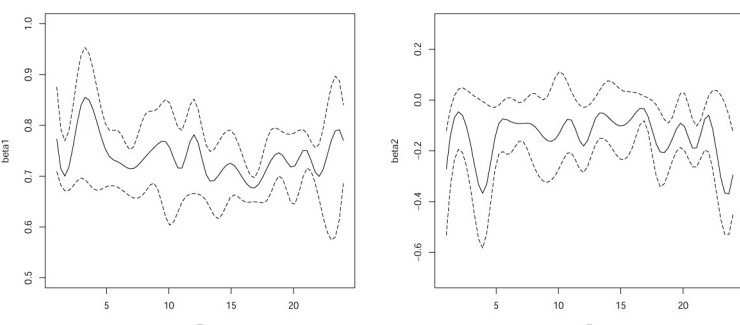

**Fig 4. Estimates and the corresponding pointwise confidence interval of $\beta 1(t)$, $\beta 2(t)$ for interval censored data.**

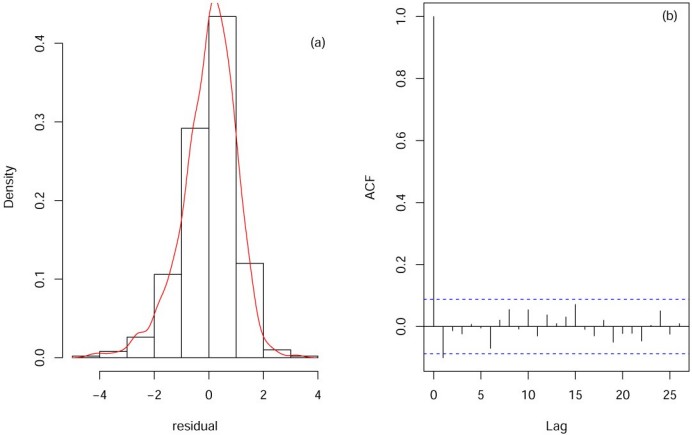

**Fig 5. Residual histogram (a) and AFC (b) plot.**

data. As shown in the above results, when the data cannot be fully observed, our proposed method can well estimate the coefficient function.

## Conclusions

In this paper, firstly proposes a coefficient function estimation method (DGQR estimation) for interval censored quantile regression with varying-coefficient model, which creatively solves the problem of interval censoring of response variables under the model. The property of asymptotic normality is established with a bias converging to zero and asymptotic normality are given a strict proof. We proposed methods do not require the interval censoring vectors to be identically distributed, and can be applied to models with fixed, discrete random, or continuous random design covariates. An other important advantage of the proposed methods is their computational simplicity, and all objective functions of the minimization problems involved in the proposed methods are simple, convex, and easy to treat. In the simulation, we put in the Uniform kernel, our simulation results support the validity of our methods. Finally, a real data sets analysis show that intervel censored of quantile regression with varying-coefficient model for the air pollution data set. The empirical analysis results are significant. Therefore the DGQR estimation for interval censored quantile regression with varying-coefficient models can be applied to alleviate the curse of dimensionality application.

## Appendix

Nothing that $F_\tau(x_i^\top[a + b(T_i - t)])$ is free of $a$ and the minimization in problem (3) is taken over $a$, we rewrite problem (3) in the following:

$$\min \sum_{i=1}^{n} \left\{ F_\tau\left(t_{1i}, t_{2i}, x_i^\top[a + b(T_i - t)]\right) - F_\tau\left(t_{1i}, t_{2i}, x_i^\top[\beta(t) + \beta'(t)(T_i - t)]\right) \right\} K\left(\frac{T_i - t}{h}\right).$$

In order to prove the theorem, we establish the following four lemmas under the assumption $C.1$–$C.8$ for any $\tau \in (0, 1)$.

**Lemma 1**. If $S(u_1, u_2) = (1 - \tau)|t_2 - \max(t_2, u_2)| + \tau|t_1 - \min(t_1, u_2)| - (1 - \tau)|t_2 - \max(t_2, u_1)| - \tau|t_1 - \min(t_1, u_1)|$, $u_2 = u_1 + a$, $t_1 < t_2$, $P(t_1 < u_1 < t_2) \to 0$, and define $t_1$ and $t_2$ cannot

belong to $\Lambda = [u_1, u_2]$ at the same time, then we can obtian

$$
\begin{aligned}
S(u_1, u_2) &= S(u_1, u_2)\mathrm{sgn}(a)[I(t_2 \in \Lambda) + I(t_1 \in \Lambda) + I(t_1, t_2 \notin \Lambda)] \\
&= A_1 + A_2 + A_3 + A_4 + A_5,
\end{aligned}
$$

where

$$
\mathrm{sgn}(a) = \begin{cases} 1, & a > 0; \\ -1, & a < 0. \end{cases}
$$

$$
A_1 = (1 - \tau)(u_1 - t_2)\mathrm{sgn}(a)I(t_2 \in \Lambda),
$$
$$
A_2 = \tau \cdot (u_1 - t_1)\mathrm{sgn}(a)I(t_1 \in \Lambda),
$$
$$
A_3 = (1 - \tau) \cdot a \cdot \mathrm{sgn}(a)I(t_2 \in \Lambda),
$$
$$
A_4 = \tau \cdot a \cdot \mathrm{sgn}(a)I(t_1 \in \Lambda),
$$
$$
A_5 = a \cdot [(1 - \tau)I(u_1 > t_2) - \tau I(u_1 \leq t_1)].
$$

**Lemma 2**.

$$
ES_n(z(t)) = \frac{1}{2}f_T(t)z^\top(t)z(t) + z^\top(t)H_n^{-1/2}L_n + o(1),
$$

holds uniformly in $n$ and uniformly over $\|z(t)\| \leq v$ with $v \to 0$.
where

$$
z(t) = H_n^{1/2}\sqrt{nh}[(a - \beta(t))^\top, (b - \beta'(t))^\top h]^\top,
$$

$$
L_n = \frac{1}{\sqrt{nh}}\sum_{i=1}^{n} E\{x_i^*[(1 - \tau)I(x_i^\top \beta(T_i) - r_i(t) > t_{2i}) - \tau I(x_i^\top \beta(T_i) - r_i(t) \leq t_{1i})]K_i(t)\},
$$

$$
K_i(t) = K\left(\frac{T_i - t}{h}\right).
$$

**Proof Lemma 2**. We provide $F_\tau(t_{1i}, t_{2i}, x_i^\top[a + b(T_i - t))$ as

$$
\begin{aligned}
F_\tau(t_{1i}, t_{2i}, x_i^\top[a + b(T_i - t)) &= (1 - \tau)|t_{2i} - \max(t_{2i}, x_i^\top[a + b(T_i - t)])| \\
&\quad + \tau|t_{1i} - \min(t_{1i}, x_i^\top[a + b(T_i - t)])|.
\end{aligned}
$$

Hence if we let

$$
r_i(t) = x_i^\top[\beta(T_i) - \beta(t) - \beta'(t)(T_i - t)],
$$

$$
x_{ni}^* = H_n^{-1/2}x_i^*,
$$

$$
\Delta_i(t) = x_i^* H_n^{-1/2}z(t)/\sqrt{nh} = z^\top(t)x_{ni}^*/\sqrt{nh}.
$$

We can decompose

$$
\begin{aligned}
x_i^\top[a + b(T_i - t)] &= x_i^\top\beta(T_i) + x_i^\top[a - \beta(t)] + x_i^\top(T_i - t)[b - \beta'(t)] - r_i(t) \\
&= x_i^\top\beta(T_i) - r_i(t) + \Delta_i(t), \\
x_i^\top[\beta(t) + \beta'(t)(T_i - t)] &= x_i^\top\beta(T_i) - r_i(t).
\end{aligned}
$$

Then we have rewrite $S_n(a, b, t)$ as $S_n(z(t))$

$$S_n(z(t)) = \sum_{i=1}^{n} \left\{ F_\tau[t_{1i}, t_{2i}, x_i^\top \beta(T_i) - r_i(t) + z^\top(t)x_{ni}^*/\sqrt{nh}] - F_\tau[t_{1i}, t_{2i}, x_i^\top \beta(T_i) - r_i(t)] \right\} K_i(t).$$

For notational convenience, let

$$f_{ni}(z(t)) = \left\{ F_\tau[t_{1i}, t_{2i}, x_i^\top \beta(T_i) - r_i(t) + z^\top(t)x_{ni}^*/\sqrt{nh}] - F_\tau[t_{1i}, t_{2i}, x_i^\top \beta(T_i) - r_i(t)] \right\} K_i(t).$$

By Lemma 1 rewrite $x_i^\top \beta(T_i) - r_i(t))$ as $u_1$, $x_i^\top \beta(T_i) - r_i(t) + Z^\top(t)x_{ni}^*/\sqrt{nh}$ as $u_2$, $z^\top(t)x_{ni}^*/\sqrt{nh}$ as $a$, than let $\Lambda_i$ be the interval with $x_i^\top \beta(T_i) - r_i(t))$ and $x_i^\top \beta(T_i) - r_i(t) + z^\top(t)x_{ni}^*/\sqrt{nh}$ as two end point, thus

$$S_n(z) = I_1 + I_2 + I_3 + I_4 + I_5,$$

where

$$I_1 = (1-\tau)\sum_{i=1}^{n}[(x_i^\top \beta(T_i) - r_i(t) - t_{2i})\text{sgn}(z^\top(t)x_{ni}^*/\sqrt{nh})\text{I}(t_{2i}\epsilon\Lambda_i)K_i(t)],$$

$$I_2 = \tau\sum_{i=1}^{n}[(x_i^\top \beta(T_i) - r_i(t) - t_{1i})\text{sgn}(z^\top(t)x_{ni}^*/\sqrt{nh})\text{I}(t_{1i}\epsilon\Lambda_i)K_i(t)],$$

$$I_3 = (1-\tau)\sum_{i=1}^{n}[(z^\top(t)x_{ni}^*/\sqrt{nh})\text{sgn}(z^\top(t)x_{ni}^*/\sqrt{nh})\text{I}(t_{2i}\epsilon\Lambda_i)K_i(t)],$$

$$I_4 = \tau\sum_{i=1}^{n}[(z^\top(t)x_{ni}^*/\sqrt{nh})\text{sgn}(z^\top(t)x_{ni}^*/\sqrt{nh})\text{I}(t_{1i}\epsilon\Lambda_i)K_i(t)],$$

$$I_5 = \sum_{i=1}^{n}K_i(t)(z^\top(t)x_{ni}^*/\sqrt{nh})[(1-\tau)I(x_i^\top \beta(T_i) - r_i(t) > t_{2i}) - \tau I(x_i^\top \beta(T_i) - r_i(t) \le t_{1i})].$$

Noting that $P(t_{1i} < t_{2i}) = 1$, by Assumptions C.1–C.8, it is also easy to show that

$$E(I_1) = E[E(I_1|T_i)] = (1-\tau)\sum_{i=1}^{n}E\{K_i(t)E[(x_i^\top \beta(T_i) - r_i(t) - t_{2i})I(t_{2i}\epsilon\Lambda_i)|T_i]\}.$$

Using mean value theorems for definite integrals, we have

$$E[(x_i^\top \beta(T_i) - r_i(t) - t_{2i})I(t_{2i} \in \Lambda_i)|T_i] = f(\mu)\left(z^\top(t)x_{ni}^*/\sqrt{nh}\right)\frac{\partial G_i^2(\ell)}{\partial \ell}|_{\ell = x_i^\top \beta(T_i)} + o(1),$$

where $f(\mu) = x_i^\top \beta(T_i) - r_i(t) - \mu$. By a Taylor expansion, $f(\mu) = -\frac{1}{2}\left(z^\top(t)x_{ni}^*/\sqrt{nh}\right) + o(1)$. Thus, we can obtain

$$E(I_1) = -\frac{1}{2}(1-\tau)f_T(t)z^\top(t)\left\{\frac{1}{n}\sum_{i=1}^{n}Ex_{ni}^* x_{ni}^{*\top}\frac{\partial G_i^2(\ell)}{\partial \ell}|_{\ell = x_i^\top \beta(T_i)}\right\}z(t) + o(1).$$

Imitating the calculation process of $E(I_1)$, we have

$$E(I_2) \quad = E[E(I_2|T_i)] = -\frac{1}{2}\tau f_T(t)z^\top(t)\left\{\frac{1}{n}\sum_{i=1}^n Ex_{ni}^* x_{ni}^{*\top}\frac{\partial G_i^1(r)}{\partial r}\Big|_{r=x_i^\top \beta(T_i)}\right\}z(t) + o(1),$$

$$E(I_3) \quad = E[E(I_3|T_i)] = (1-\tau)f_T(t)z^\top(t)\left\{\frac{1}{n}\sum_{i=1}^n Ex_{ni}^* x_{ni}^{*\top}\frac{\partial G_2^i(\ell)}{\partial \ell}\Big|_{\ell=x_i^\top \beta(T_i)}\right\}z(t) + o(1),$$

$$E(I_4) \quad = E[E(I_4|T_i)] = \tau f_T(t)z^\top(t)\left\{\frac{1}{n}\sum_{i=1}^n Ex_{ni}^* x_{ni}^{*\top}\frac{\partial G_1^i(r)}{\partial r}\Big|_{r=x_i^\top \beta(T_i)}\right\}z(t) + o(1).$$

Obviously, $E(I_5) = z^\top(t)H_n^{-1/2}L_n$ hold true, where

$$L_n = \quad \frac{1}{\sqrt{nh}}\sum_{i=1}^n E\left\{x_i^*\left[(1-\tau)I\left(x_i^\top \beta(T_i) - r_i(t) > t_{2i}\right) - \tau I(x_i^\top \beta(T_i) - r_i(t) \le t_{1i})\right]K_i(t)\right\}.$$

Based on the above result, we have

$$ES_n \quad = EI_1 + EI_2 + EI_3 + EI_4 + EI_5$$
$$= \frac{1}{2}f_T(t)z^\top(t)z(t) + z^\top(t)H_n^{-1/2}L_n + o(1),$$

holds uniformly in $n$ and uniformly over $\|z(t)\| < v$ with $v \to 0$. This complete the proof of Lemma 2.

Define

$$\Delta_{ni} = [(1-\tau)I(x_i^\top \beta(T_i) - r_i(t) > t_{2i}) - \tau I(x_i^\top \beta(T_i) - r_i(t) \le t_{1i})]x_{ni}^* K_i(t),$$

which is the derivative of $f_{ni}(z(t))$ at $z(t) = 0$ expect $x_i^\top \beta(T_i) - r_i(t) = t_{1i}$ or $x_i^\top \beta(T_i) - r_i(t) = t_{2i}$.

**Lemma 3**. Let $R_{ni}(z(t)) = f_{ni}(z(t)) - (nh)^{-1}\Delta_{ni}^\top z(t)$. Then

$$|R_{ni}(z(t))| \quad \le |z^\top(t)x_{ni}^*/\sqrt{nh}|[(1-\tau)I(|t_{2i} - x_i^\top \beta(T_i) + r_i(t)| < |z^\top(t)x_{ni}^*/\sqrt{nh}|)$$
$$+ \tau I(|t_{1i} - x_i^\top \beta(T_i) + r_i(t)| < |z^\top(t)x_{ni}^*/\sqrt{nh}|)]K_i(t).$$

**Proof of Lemma 3**. It follows directly from Lemma 2 in [17].

**Lemma 4**. For any $\tau \in (0, 1)$

$$\sup_{Z \in \emptyset} |S_n(z(t)) - ES_n(z(t))| = o_p(1),$$

holds for any bounded subset $\emptyset \in \Re^m$ as $n \to \infty$;

$$-\frac{1}{n}\sum_{i=1}^n [R_{ni}(z(t)) - ER_{ni}(z(t))] = o_p\left(\frac{\|z(t)\|}{\sqrt{nh}}\right),$$

holds uniformly in $n$ and uniformly over $0 < \|z(t)\| < Z$ as $v \to 0$.

**Proof of Lemma 4**. It follows directly from Lemma 3 in [17].

**Proof of Theoren 1**: Note that

$$\|L_n\|^2 \leq \frac{1}{\sqrt{nh}}\sum_{i=1}^{n}\|x_i^* K_i(t)\|^2 [(1-\tau)P(x_i^\top \beta(T_i) - r_i(t) > t_{2i})$$
$$-\tau P(x_i^\top \beta(T_i) - r_i(t) \leq t_{1i})]^2,$$

holds for $n$ large enough. By the fact $P(t_{1i} < y \leq t_{2i}) = 1$, we have

$$P(t_{2i} < x_i^\top \beta(T_i) - r_i(t)) = P(y_i < x_i^\top \beta(T_i) - r_i(t)) - P(y_i < x_i^\top \beta(T_i) - r_i(t) \leq t_{2i})),$$

$$P(t_{1i} \geq x_i^\top \beta(T_i) - r_i(t))) = P(y_i > x_i^\top \beta(T_i) - r_i(t))) - P(y_i > x_i^\top \beta(T_i) - r_i(t) > t_{1i}))).$$

By Assumption $C.1–C.8$, we can get the following results

$$[(1-\tau)P(x_i^\top \beta(T_i) - r_i(t) > t_{2i}) - \tau P(x_i^\top \beta(T_i) - r_i(t) > t_{1i}]^2$$
$$\leq [P(t_{1i} < x_i^\top \beta(T_i) - r_i(t) \leq t_i + \varrho_n)]^2$$
$$\leq [P(t_{1i} - \varrho_n < x_i^\top \beta(T_i) - r_i(t) \leq t_i + \varrho_n)]^2$$
$$= [P(|t_{1i} - x_i^\top \beta(T_i) + r_i(t)| \leq +\varrho_n)]^2$$
$$= O(\varrho_n^2).$$

Under the Assumption C.8 we know $\|L_n\|^2 = O(\varrho_n^2)$, $\|L_n\| \mapsto^P 0$, and we know $H_n$ is bounded away from zero for $n$ large enough. Then we show that for any $v > 0$, $\|H_n^{-1/2}L_n\| < \frac{v}{4}$, holds for all $n$ large enough, and $v$ small enough.

By Lemma 2 we know for any $v > 0$ small enough, there is $\epsilon > 0$ such that

$$ES_n(z(t)) \geq \frac{1}{2}f_T(t)v^2 - v \cdot \frac{v}{4} + o(v^2) \geq \epsilon,$$

holds for any $\|z(t)\| = v$ and $n$ large enough. By Lemma 3 we have that for any $\delta > 0$,

$$(1-\delta) \leq P\left(\sup_{\|z(t)\|<v} |S_n(z(t)) - ES_n(z(t))| < \frac{\epsilon}{2}\right)$$
$$\leq P\left(\sup_{\|z(t)\|=v} |S_n(z(t)) - ES_n(z(t))| < \frac{\epsilon}{2}\right)$$
$$\leq P\left(\inf_{\|z(t)\|=v} S_n(z(t)) \geq \frac{\epsilon}{2}\right),$$

holds for any $n$ large enough. Nothing that $S_n(z(t))$ is convex and $S_n(0) = 0$, we can conclude that $\|\hat{z}_n\| < v$ holds true with probability tending to 1 as $n \to \infty$.

**Proof of Theorem 2.** Let $W_n = \frac{1}{\sqrt{nh}}\sum_{i=1}^{n}\Delta_{ni}$, and $S_n(z(t)) = \sum_{i=1}^{n} f_{ni}(z(t))$.

$$S_n(\hat{z}_n(t)) = ES_n(\hat{z}_n(t)) + (nh)^{-1/2}W_n^\top \hat{z}_n(t) - E(nh)^{-1/2}W_n^\top \hat{z}_n(t)$$
$$+ [S_n(\hat{z}_n(t)) - ES_n(\hat{z}_n(t) - (nh)^{-1/2}W_n^\top \hat{z}_n(t) + E(nh)^{-1/2}W_n^\top \hat{z}_n(t)],$$

where

$$S_n(\hat{z}_n(t)) - ES_n(\hat{z}_n(t) - (nh)^{-1/2}W_n^\top\hat{z}_n(t) + E(nh)^{-1/2}W_n^\top\hat{z}_n(t)$$

$$= \sum_{i=1}^{n}[R_{ni}(\hat{z}_n(t)) - ER_{ni}(\hat{z}_n(t))] \to o_p\left(\frac{\|\hat{z}_n(t)\|}{\sqrt{nh}}\right).$$

According to above conclusions and Lemma 2 we have

$$S_n(\hat{z}_n(t)) \quad = \frac{1}{2}f_T(t)\hat{z}_n^\top(t)\hat{z}_n(t) + \hat{z}_n^\top(t)H_n^{-1/2}L_n + (nh)^{-1/2}W_n^\top\hat{z}_n(t)$$

$$-E\left[(nh)^{-1/2}W_n^\top\hat{z}_n(t)\right] + o_p\left(\frac{\|\hat{z}_n(t)\|}{\sqrt{nh}}\right) + o(1).$$

Since $\hat{z}_n(t)$ is the minimization point of $S_n(\hat{z}_n(t))$, then

$$\frac{\partial S_n(\hat{z}_n(t))}{\partial \hat{z}_n(t)} = f_T(t)\hat{z}_n(t) + H_n^{-1/2}L_n + (nh)^{-1/2}W_n^\top - E\left[(nh)^{-1/2}W_n^\top\right].$$

Let $\frac{\partial S_n(\hat{z}_n(t))}{\partial \hat{z}_n(t)} = 0$, by direct calculation we know

$$\hat{z}_n(t) = -\frac{H_n^{-1/2}L_n + (nh)^{-1/2}(W_n^\top - EW_n^\top) + o_p((nh)^{-1/2})}{f_T(t) + o(1)},$$

then

$$\sqrt{nh}[(\hat{a} - \beta(t))^\top, (\hat{b} - \beta'(t))^\top h]^\top$$

$$= -\frac{1}{f_T(t)}\left\{H_n^{-1}L_n + (nh)^{-1/2}H_n^{-1/2}W_n^\top - E\left[(nh)^{-1/2}H_n^{-1/2}W_n^\top\right]\right\},$$

where

$$W_n^\top = \quad \frac{1}{\sqrt{nh}}\sum_{i=1}^{n}[(1-\tau)I(x_i^\top\beta(T_i) - r_i(t) > t_{2i}) - \tau I(x_i^\top\beta(T_i) - r_i(t) \le t_{1i})]H_n^{-1/2}x_i^*K_i(t).$$

Let $C_i = [(1-\tau)I(x_i^\top\beta(T_i) - r_i(t) > t_{2i}) - \tau I(x_i^\top\beta(T_i) - r_i(t) \le t_{1i})]$, then $W_n^\top = \frac{1}{\sqrt{nh}}\sum_{i=1}^{n}C_iH_n^{-1/2}x_i^*K_i(t)$. Thus

$$\sqrt{nh}H_nf_T(t)((\hat{a} - \beta(t))^\top, (\hat{b} - \beta'(t))^\top h)^\top + L_n$$

$$= \quad (nh)^{-1}E\sum_{i=1}^{n}C_ix_i^*K_i(t) - (nh)^{-1}\sum_{i=1}^{n}C_ix_i^*K_i(t).$$

Then calculate the variance of $(nh)^{-1}\sum_{i=1}^{n}C_ix_i^*K_i(t)$,

$$\mathrm{Var}\left((nh)^{-1}\sum_{i=1}^{n}C_ix_i^*K_i(t)\right)$$

$$= (nh)^{-1}\sum_{i=1}^{n}\left\{E(C_i^2x_i^*K_i^2(t)x_i^{*\top}) - E[C_ix_i^*K_i(t)]E[C_ix_i^*K_i(t)]\right\}$$

$$= (nh)^{-1}\sum_{i=1}^{n}E\left\{x_i^*x_i^{*\top}[(1-\tau)^2P_{2i} + \tau^2P_{1i} + 2\tau(\tau - 1)P_i]f_T(t)\right\} + o_p(h)$$

$$= \tilde{H}_n + o_p(h),$$

where

$$P_{1i} = P(x_i^\top \beta(T_i) \le t_{1i}|X_i, T_i)P(x_i^\top \beta(T_i) > t_{1i}|X_i, T_i) + o(1),$$

$$P_{2i} = P(x_i^\top \beta(T_i) > t_{2i}|X_i, T_i)P(x_i^\top \beta(T_i) \le t_{2i}|X_i, T_i) + o(1),$$

$$P_i = P(x_i^\top \beta(T_i) > t_{2i}|X_i, T_i)P(x_i^\top \beta(T_i) \le t_{1i}|X_i, T_i) + o(1),$$

$$\tilde{H}_n = (nh)^{-1} \sum_{i=1}^n E\left\{x_i^* x_i^{*\top}[(1-\tau)^2 P_{2i} + \tau^2 P_{1i} + 2\tau(\tau-1)P_i]f_T(t)\right\}.$$

Noting the fact that $(nh)^{-1}\sum_{i=1}^n C_i x_i^* K_i(t) - E(nh)^{-1}\sum_{i=1}^n C_i x_i^* K_i(t) \sim N(0, \tilde{H}_n)$. Then

$$\sqrt{nh}\tilde{H}_n^{-1/2}H_n f_T(t)((\hat{a}-\beta(t))^\top, (\hat{b}-\beta'(t))^\top h)^\top + \tilde{H}_n^{-1/2}L_n \xrightarrow{d} N(0, E_m).$$

Therefore, we have Theorem 2 holds true.

## Supporting information

**S1 File.**
(ZIP)

## Author Contributions

**Software:** Yun Li.

**Writing – original draft:** ChunJing Li, Yun Li, Xue Ding.

**Writing – review & editing:** XiaoGang Dong.

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
