## [Decision Letter · Decision Letter 0]

18 Sep 2020

DGQR estimation for interval censored quantile regression with varying-coefficient models

PONE-D-20-18653

Dear Dr. Dong,

We’re pleased to inform you that your manuscript has been judged scientifically suitable for publication and will be formally accepted for publication once it meets all outstanding technical requirements.

Kind regards,

Maria Alessandra Ragusa, PhD Professor

Academic Editor

PLOS ONE

1.

Additional Editor Comments (optional):

The second revision follows all the suggestions, then the paper is now ready for publication.

Reviewers' comments:

Reviewer's Responses to Questions

**Comments to the Author**

1. Is the manuscript technically sound, and do the data support the conclusions?

Reviewer #1: Yes

2. Has the statistical analysis been performed appropriately and rigorously? 

Reviewer #1: Yes

3. Have the authors made all data underlying the findings in their manuscript fully available?

Reviewer #1: Yes

4. Is the manuscript presented in an intelligible fashion and written in standard English?

Reviewer #1: Yes

5. Review Comments to the Author

Reviewer #1: In the manuscript "DGQR estimation for interval censored quantile regression with varying-coefficient models", a coefficient function estimation method (DGQR estimation) for interval censored quantile regression with varying-coefficient model, which creatively solves the problem of interval censoring of response variables under the model, was proposed. After examining the text, my consideration is that the research was conducted following a technical standard high enough to allow solid conclusions to be drawn from the results. The methods are described in detail and the existence of solutions to the problem under consideration is a relevant objective.

The authors present a detailed and well-organized explanation of the study. The structure of the document, the reconstruction of the preliminary studies and the methodology, are rigorously explained and well-performed, and examples and simulations were given.

The document is certainly worthy of publication in " PLOS ONE", considering the suggested minor changes.

• Abstract, Line 1: “proposes” in […This paper propose a direct generalization…];

• Page 1, Line 5: “first” instead of […were frist introduced …];

• Page 8, Line 41: “ an estimate” in […to develop a estimate…];

• Page 8, Line 57: “is given” in […A conclusion are given in …];

• Page 8, Line 58: insert dot at the end of the sentence. […proofs are given].

6. PLOS authors have the option to publish the peer review history of their article (what does this mean?). If published, this will include your full peer review and any attached files.

Reviewer #1: No

---

## [Editor Report · Acceptance letter]

27 Oct 2020

PONE-D-20-18653 

DGQR estimation for interval censored quantile regression with varying-coefficient models 

Dear Dr. Dong:

I'm pleased to inform you that your manuscript has been deemed suitable for publication in PLOS ONE. Congratulations! Your manuscript is now with our production department. 

Kind regards, 

on behalf of

Dr. Maria Alessandra Ragusa 

Academic Editor

PLOS ONE